optics/plant science

Vis/NIR reflectance spectroscopy, nitrogen fertilizer level, chlorophyll content, support vector machine, partial least square

**Author for correspondence:**
Jiandong Hu
e-mail: jdhu@henau.edu.cn

# Vis/NIR reflectance spectroscopy for hybrid rice variety identification and chlorophyll content evaluation for different nitrogen fertilizer levels

Hao Zhang[1,3], Zheng Duan[4], Yiyun Li[4], Guangyu Zhao[4], Shiming Zhu[4], Wei Fu[4], Ting Peng[2], Quanzhi Zhao[2], Sune Svanberg[4,5] and Jiandong Hu[1,3]

[1]College of Mechanical and Electrical Engineering, and [2]Collaborative Innovation Center of Henan Grain Crops/Key Laboratory of Rice Biology in Henan Province, Henan Agricultural University, Zhengzhou 450002, People's Republic of China
[3]Henan International Joint Laboratory of Laser Technology in Agriculture Sciences, Zhengzhou 450002, People's Republic of China
[4]Center for Optical and Electromagnetic Research, South China Normal University, Guangzhou 510006, People's Republic of China
[5]Atomic Physics Division, Department of Physics, Lund University, PO Box 118, 221 00 Lund, Sweden

HZ, 0000-0002-7600-3967; JH, 0000-0003-1325-7334

Nitrogen is one of the most important nutrient indicators for the growth of crops, and is closely related to the chlorophyll content of leaves and thus influences the photosynthetic ability of the crops. In this study, five hybrid rice varieties were cultivated during one entire growing period in one experimental field supplied with six nitrogen fertilizer levels. Visible and near infrared (vis/NIR) reflectance spectroscopy combined with multivariate analysis was used to identify hybrid rice varieties and nitrogen fertilizer levels, as well as to detect chlorophyll content associated with nitrogen levels. The support vector machine (SVM) algorithm was applied to identify five varieties of hybrid rice and six levels of nitrogen fertilizer. The results demonstrated that different varieties of hybrid rice for each nitrogen level can be well distinguished except for the highest nitrogen level, and no nitrogen level for each rice variety can be completely identified from the other five nitrogen levels. Further, 12 spectral indices combined with partial least square (PLS) analysis were

applied for estimating chlorophyll content of rice leaves from plants subjected to different nitrogen levels, and a root mean square error of cross-validation (RMSECV) of 0.506, a coefficient of determination ($R^2$) of 97.8% and a ratio of performance to deviation (RPD) of 4.6 for all rice varieties indicated this as a preferable procedure. This study demonstrates that Vis/NIR spectroscopy can have a great potential for identification of rice varieties and evaluation of nitrogen fertilizer levels.

## 1. Introduction

Paddy rice dominates overall crop production on the global scale, and China is the world's largest producer of rice. To achieve high-yielding rice and solve the problems associated with food shortage in developing countries, hybrid rice was first developed by L.P. Yuan in China and has been investigated for more than 50 years. In recent years, various varieties of super hybrid rice were bred, such as ChaoYou 1000, which in 2018 has set a new yield record of about 1065 kg acre$^{-1}$ (approx. 73.5 kg ha$^{-1}$) in a test field in Sanya City, Hainan Province, China [1]. Chlorophyll is a primary pigment used in photosynthesis. The leaf chlorophyll content (LCC) is usually regarded as one main indicator to estimate the photosynthetic ability, health status and nitrogen nutrition level of crops. It is well known that nitrogen is one of the necessary elements for the growth of crops. The leaf nitrogen content (LNC) also plays a key role in indicating the photosynthetic status of crops. Previous studies have demonstrated that LNC is strongly related to LCC [2–4]. Therefore, monitoring the chlorophyll content can provide important information on the photosynthesis and nitrogen nutrition status of crops.

Spectral signals are associated with biochemical and structural properties of crops; therefore, reflectance, transmission and fluorescence are the three commonly used spectroscopic methods for non-destructive estimation of chlorophyll and nitrogen content. The soil plant analysis development (SPAD) instrument has been used as a rapid, low-cost and popular tool to estimate the relative content of chlorophyll by measuring leaf transmission signals. Numerous studies have reported the use of the SPAD meter to evaluate chlorophyll content and nitrogen status [5–8]. Visible and near infrared (vis/NIR) spectroscopy, hyperspectral light detection and ranging (LIDAR), and hyperspectral imaging have been successfully applied for reflectance measurements of crops [9–11]. Based on the combinations of spectral reflectance from two or more wavelengths, many spectral indices (SIs) have been proposed to analyse the biochemical parameters of crops, such as LCC, LNC and leaf water content (LWC) [12–14]. Related studies have demonstrated that the SIs derived from reflectance spectra in the vis/NIR region are closely related to LCC and LNC [13,15]. Laser-induced fluorescence (LIF) has also been widely applied to study the photosynthetic activity and nutrient status of crops, where the ratio of chlorophyll fluorescence emitted at 685 and 740 nm was frequently used as a key indicator [16–18]. Moreover, some other empirical leaf fluorescence parameters related to the fluorescence intensity at 460, 525, 685 and 740 nm were also demonstrated to be positively and linearly correlated with the LNC of crops [19–21].

In recent years, several studies have been pursued to accurately estimate LCC and LNC of paddy rice. These studies include the spectroscopic methods such as LIF, the combination of LIF and reflectance spectroscopy, hyperspectral LIDAR, and hyperspectral imaging [13,19,21–24]. It is well known that nitrogen over-fertilization or under-fertilization not only influences crop growth and production, but also results in serious environmental problems. Thus, it is very important to study the influence of different nitrogen fertilizer levels on the photosynthetic ability of paddy rice. Accurate identification of nitrogen fertilizer levels of paddy rice has been studied by using LIF technique combined with multivariate methods such as principle component analysis (PCA), Fisher's discriminate analysis (FDA) and support vector machine (SVM) [25–27]. However, reflectance spectroscopy has rarely been employed to analyse the photosynthetic activity of paddy rice treated with different nitrogen fertilization levels. Compared to the LIF technique, reflectance spectroscopy has the advantages including the use of compact and cheap instrumentation. Therefore, the main objectives of the present study are as follows: (i) to identify the hybrid rice varieties and the nitrogen fertilization levels by the combination of vis/NIR reflectance spectroscopy and SVM method; (ii) to analyse the performance of selected SIs and partial least square (PLS) method for the estimation of chlorophyll content (or SPAD value); and (iii) to analyse the vertical distribution of chlorophyll content in individual rice plants.

**Table 1.** Hybrid rice varieties and their parent cultivars.

| sample name | male parent | female parent |
| --- | --- | --- |
| LiangYouPeiJiu (LYPJ) | R9311 | PA64S |
| YLiangYou 1 (YLY1) | R9311 | Y58S |
| YLiangYou 2 (YLY2) | YuanHui2 | Y58S |
| YLiangYou 900 (YLY900) | R900 | Y58S |
| ChaoYou 1000 (CY1000) | R900 | GuangXiang24S |

# 2. Material and methods

## 2.1. Materials

Five varieties of hybrid rice, namely, LiangYouPeiJiu (LYPJ), YLiangYou 1 (YLY1), YLiangYou 2 (YLY2), YLiangYou 900 (YLY900) and ChaoYou 1000 (CY1000) were cultivated in 2016 in one experimental field of Xinyang City, Henan Province, China (32°28′ N to 32°29′ N, 114°02′ E to 114°03′ E). The parent cultivars of these five varieties of hybrid rice are listed in table 1. Xinyang is located in the transition zone between the northern subtropical to warm temperate zones, with climatic conditions characterized by sufficient summer sunlight, high temperature and abundant rain, which constitute ideal conditions for rice planting. The soil in the experimental field is a typical paddy soil for farming in China. The fundamental nutrients of paddy soil consist of $0.91\,\mathrm{g\,kg^{-1}}$ total nitrogen, $14.4\,\mathrm{mg\,kg^{-1}}$ available phosphorus, $12.9\,\mathrm{mg\,kg^{-1}}$ available potassium and $28.5\,\mathrm{g\,kg^{-1}}$ organic matter. Rice varieties were seeded on 12–15 April 2016, and then the seedlings were transplanted into the experimental field fertilized with fundamental nutrients on 13 May 2016. To ensure that plants from different varieties have equal density, the seedlings were manually planted in an area of $20 \times 30$ cm, and each hole on the experimental field contained two seedlings. During the entire rice growing period, six levels of nitrogen fertilizer (N1: $0\,\mathrm{kg\,ha^{-1}}$, N2: $150\,\mathrm{kg\,ha^{-1}}$, N3: $210\,\mathrm{kg\,ha^{-1}}$, N4: $300\,\mathrm{kg\,ha^{-1}}$, N5: $390\,\mathrm{kg\,ha^{-1}}$ and N6: $450\,\mathrm{kg\,ha^{-1}}$) were supplied at three different times: 30% as base fertilizer on 13 April 2016, 30% as tillering fertilizer on 19 May and 26 May 2016 and 40% as panicle fertilizer at the fourth and second leaf-age (25 June–4 July 2016). Entire phosphorus was used as base fertilizer. Potassium fertilizer was administrated at two different times: 50% as base fertilizer on 13 April 2016 and 50% as spikelet-promoting fertilizer at second leaf-age (2–4 July 2016). In total, the experimental field was divided into 30 zones according to different rice varieties and nitrogen fertilizer levels, and the area of each zone was $8 \times 4.5$ m (length × width).

The spectral measurements were performed on 31 August 2016, during the rice maturity stage. Five whole plants were randomly collected from each zone. After picking from the fields, the rice plants were immediately placed into plastic pots filled with enough water inside to keep the leaves from being affected by water deficit condition. Then, four rice leaves at different heights of each plant were retrieved, with sampling from its top (flag leaf) to its bottom, which was followed by spectral reflectance measurements. Thus, the total number of leaf samples for spectral measurements was 600.

## 2.2. Spectral measurements

The spectral measurements were carried out in a newly constructed mobile laboratory [28] from South China Normal University. The mobile laboratory was positioned near the rice experimental field to facilitate the reflectance measurements. In the laboratory, a tungsten lamp with an output power of 130 mW was used as the light source, and the broadband light was guided with a 600 μm core-diameter fibre (transmitting fibre) to vertically irradiate the sample, where the distance between the fibre end facet and the sample was about 5 mm. The diffuse reflection light was captured using another fibre (collecting fibre) with core diameter of 600 μm and fixed at an angle of 45°, where the fibre end was placed at the same height as the transmitting fibre, and then detected using a portable miniature spectrometer (USB4000, Ocean Optics, Dunedin, FL). The reflectance was calculated according to the following formula:

$$R_\lambda = \frac{R_{\mathrm{sample}}(\lambda) - D(\lambda)}{R_{\mathrm{reference}}(\lambda) - D(\lambda)}, \tag{2.1}$$

**Table 2.** Spectral indices used in this work.

| index | formula | reference |
|---|---|---|
| $SR_{(750,550)}$ | $R_{750}/R_{550}$ | [31] |
| $SR_{(750,700)}$ | $R_{750}/R_{700}$ | [29] |
| $SR_{(800,670)}$ | $R_{800}/R_{670}$ | [30] |
| RMI | $R_{750}/R_{720} - 1$ | [32] |
| GMI | $R_{750}/R_{550} - 1$ | [32] |
| NDVI1 | $(R_{800} - R_{670})/(R_{800} + R_{670})$ | [33] |
| NDVI2 | $(R_{750} - R_{670})/(R_{750} + R_{670})$ | [34] |
| MCARI1 | $[(R_{700} - R_{670}) - 0.2 \times (R_{700} - R_{550})] \times (R_{700}/R_{670})$ | [35] |
| MCARI2 | $[(R_{700} - R_{670}) - 0.2 \times (R_{700} - R_{550})] \times (R_{750}/R_{705})$ | [36] |
| TVI | $0.5 \times [120 \times (R_{750} - R_{550}) - 200 \times (R_{670} - R_{550})]$ | [36] |
| MTVI1 | $1.2 \times [1.2 \times (R_{800} - R_{550}) - 2.5 \times (R_{670} - R_{550})]$ | [37] |
| MTVI2 | $\dfrac{1.5 \times [1.2 \times (R_{800} - R_{550}) - 2.5 \times (R_{670} - R_{550})]}{\sqrt{(2 \times R_{800} + 1)^2 - (6 \times R_{800} - 5 \times \sqrt{R_{670}}) - 0.5}}$ | [37] |

where $R_{\text{sample}}(\lambda)$ is the recorded intensity at the wavelength $\lambda$ from the sample, $R_{\text{reference}}(\lambda)$ is the recorded intensity at the wavelength $\lambda$ from a standard reference made with polytetrafluoroethylene material (the reflectivity is more than 98% in the wavelength range of 250–2200 nm) and $D(\lambda)$ is the recorded intensity at the wavelength $\lambda$ in dark conditions. The dark reflectance spectrum was recorded when all lamps in the laboratory were turned off and the fibres were obscured with dark papers. Each leaf sample was measured three times and the recorded data were then averaged before processing.

## 2.3. Selection of spectral indices

SIs or vegetation indices have been widely used for evaluating LCC and LNC. Most developed SIs are based on spectral reflectance in the wavelength region of 400–850 nm, which corresponds to the photosynthetically active radiation region of crops [29]. Stroppiana *et al.* [30] demonstrated that the spectral bands at about 550, 670, 750 and 800 nm were good indicators for LNC detection, and Yang *et al.* [22] reported that the triangular vegetation index (TVI) and the modified triangular vegetation index (MTVI1 and MTVI2) were suitable for LNC monitoring. Based on these studies, a total of 12 spectral indices were selected to analyse the LCC of hybrid rice, as listed in table 2, consisting of three simple ratios $SR_{(750,550)}$, $SR_{(750,700)}$ and $SR_{(800,670)}$; red-edge model index (RMI); green-edge model index (GMI); normalized difference vegetation index (NDVI1 and NDVI2); modified chlorophyll absorption ratio index (MCARI1 and MCARI2); TVI; MTVI1; and MTVI2.

## 2.4. Data analysis

During spectral measurements, owing to the influence of non-ideal instruments and sample properties, the obtained reflectance spectra were very noisy, which introduced some large errors to the estimation of chlorophyll content when using the SIs. Therefore, to reduce the noise and increase the signal to noise ratio, the reflectance spectra were processed by Savitzky–Golay (SG) smoothing method, with a third-order polynomial approximation and a window size of 10 points.

For each nitrogen fertilizer level, discrimination analysis for five varieties of hybrid rice was conducted by using an SVM model based on the radical basis function (RBF) approach. The penalty parameter C and kernel function parameter $\gamma$ play a very important role in controlling the modelling complexity and classification accuracy of the SVM model. Therefore, in this study, genetic algorithm (GA) was used for appropriate selection of parameters C and $\gamma$ to obtain an optimized SVM model, where the parameters used for the GA were set as: maximum generation of 100, population size of 20 and crossover probability of 0.9. The best C and $\gamma$ were determined according to the best classification accuracy based on K-fold cross-validation (K-CV). To evaluate the performance of the established

SVM discrimination model, the recognition rate (RCR) and rejection rate (RJR) were proposed to reflect the reliability of clustering among different varieties, as described in a literature study [38]. The RCR refers to the ratio of the number of samples identified by a certain variety to the total number of samples from the variety, while the RJR refers to the ratio of the number of samples rejected by a certain variety to the total number of samples from other varieties.

Further, to estimate the LCC of hybrid rice, the PLS regression method was applied to establish the calibration model. The performance of the PLS model was evaluated by using the root mean square error of cross-validation (RMSECV), the root mean square error of prediction (RMSEP), the coefficient of determination ($R^2$) and the ratio of performance to deviation (RPD). The RMSECV, RMSEP, $R^2$ and RPD [39] are calculated by using the following expressions:

$$\text{RMSECV} = \sqrt{\frac{\sum_{i=1}^{n}(y_i - \hat{y}_i)^2}{n}}, \quad (2.2)$$

$$\text{RMSEP} = \sqrt{\frac{\sum_{j=1}^{m}(y_j - \hat{y}_j)^2}{m}}, \quad (2.3)$$

$$R^2 = 1 - \frac{\sum_{k=1}^{N}(\hat{y}_k - y_k)^2}{\sum_{k=1}^{N}(y_k - \bar{y}_k)^2} \quad (2.4)$$

and

$$\text{RPD} = \frac{\text{s.d.}}{\text{RMSE}}, \quad (2.5)$$

where $n$ and $m$ are the sample number in the training and prediction set, respectively. $N$ denotes the total sample number in the given sample set. $\hat{y}_i$, $\hat{y}_j$ and $\hat{y}_k$ are the predicted values in the training set, prediction set and given sample set, respectively. $y_i$, $y_j$ and $y_k$ represent the reference values in the training set, prediction set and given sample set, respectively. $\bar{y}_k$ is the average reference values in the given sample set. s.d. is the standard deviation of the chlorophyll content or SPAD values measured using the hand-held SPAD meter (SPAD-502Plus), and RMSE is the root mean square error obtained from the PLS model. High values of $R^2$ and RPD, as well as low values of RMSECV and RMSEP indicate a good PLS model. Moreover, an RPD value greater than 2.5 corresponds to an excellent model, an RPD value between 2.0 and 2.5 corresponds to a very good model, while an RPD value between 1.8 and 2.0 still indicates a good model [40,41].

# 3. Results and discussion

## 3.1. Spectral reflectance of hybrid rice

Figure 1a exhibits typical reflectance spectra of hybrid rice subjected to the N1 nitrogen level, in the wavelength range of 450–850 nm. After pre-treatment with SG smoothing, the reflectance spectra are shown in figure 1b. Clearly, a major valley appears at about 670 nm, which is attributed to chlorophyll absorption. In the region of 510–570 nm, a peak occurs at about 550 nm, and the abrupt reflectance change at about 530 nm is called the 'green-edge'. In the region of 690–750 nm, reflectance also changes suddenly in the vicinity of 700 nm, which is called the 'red-edge'. In fact, both the 'green-edge' and 'red-edge' positions are highly related to the chlorophyll content. The edge gets shifted toward longer wavelength with the increase of chlorophyll content; and this phenomenon is called the 'red-shift' [31,42,43]. Therefore, these two parameters have been commonly used to estimate the chlorophyll content. Figure 1c,d demonstrates that both the 'green-edge' and 'red-edge' position present an evident 'red-shift' with increasing nitrogen levels from N1 to N4, while the change is not clear with the increase in nitrogen levels from N4 to N6, which indicates that moderate doses of nitrogen fertilizer can promote rice growth.

## 3.2. Identification of rice varieties and nitrogen levels based on support vector machine

A total of 225 reflectance spectra from five varieties of hybrid rice were used for classification. With a number ratio of 5 : 1, the spectral data were divided into a training set and a testing set, where 187 spectra among them were used as training set and the remaining 38 spectra were used as testing set. For six different nitrogen levels, the classification results of SVM model are presented in table 3, where RCR and RJR were used to describe the identification accuracy. Both the RCR and RJR values infer that LYPJ and CY1000 from the N1–N5 nitrogen levels could be clearly distinguished from the

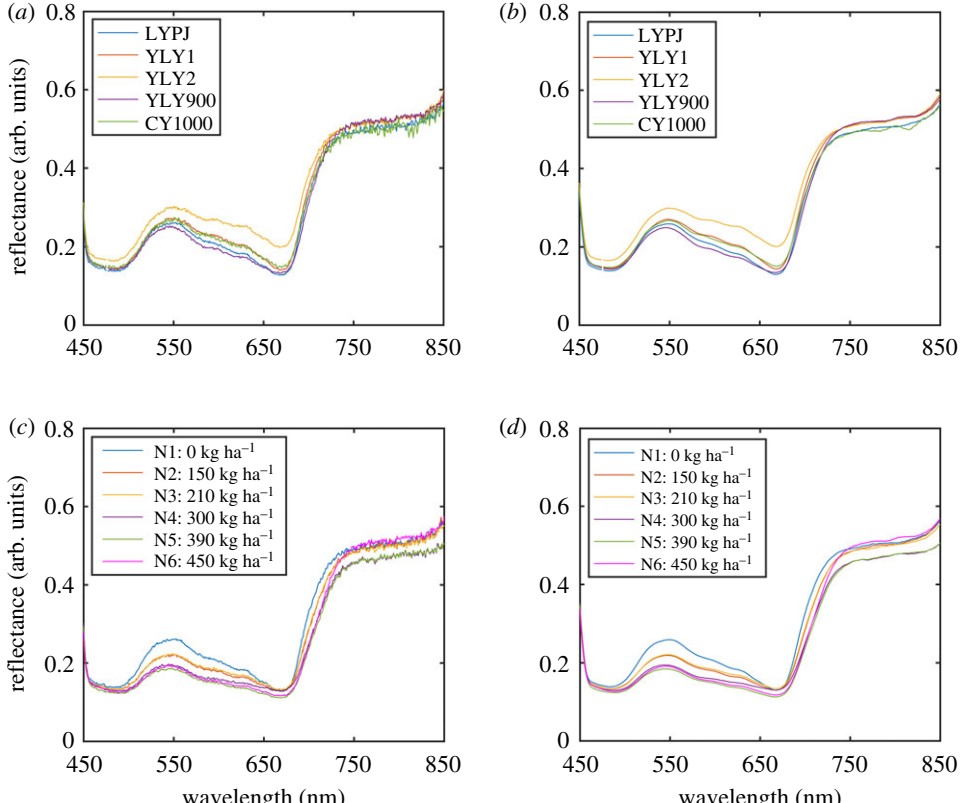

**Figure 1.** Reflectance spectra of five hybrid rice varieties without nitrogen application (N1) (*a*) before and (*b*) after preprocessing. Reflectance spectra of LYPJ hybrid rice with different nitrogen application levels before (*c*) and after preprocessing (*d*).

other three rice varieties, except that CY1000 from N4 nitrogen level had a lower recognition accuracy (37.5%). Comparatively low classification accuracy of YLY1, YLY2 and YLY900 was mainly caused by the mis-classification between each other. This phenomenon occurred mostly because of the same female parent for these three rice varieties, resulting in the fact that these three species of hybrid rice could not be clearly distinguished. However, for all the five rice varieties under the N6 nitrogen level, the classification accuracy was not satisfactory, possibly because the hybrid rice varieties were supplied with excessive nitrogen fertilizer, which resulted in the abnormal growth of hybrid rice. Although the classification of hybrid rice with excessive nitrogen level was not successful, it was also concluded that the reflectance spectroscopy can provide a potential method for identification of different varieties of rice.

Then, the SVM method was also applied to identify different levels of nitrogen fertilizer. A total of 270 reflectance spectra from each variety of hybrid rice were used for classification. Similarly, with a number ratio of 5 : 1, the spectral data were divided into a training set and a testing set, where 225 spectra among them were used as training set and the remaining 45 spectra were used as testing set. For each rice variety, the results of the testing set are listed in table 4. Obviously, N1 (no nitrogen fertilizer applied) can be distinguished from the other five nitrogen levels for all five rice varieties, with an average recognition rate up to 90%. In contrast, N6 presents an average recognition rate of 71.4%, which is higher than those of N2–N5, and N4 shows the lowest recognition accuracy (34.3%). The lower average rejection rate of N5 and N6 indicates the mis-identification of several samples from these two N levels, which may indicate an excessive nitrogen application. Furthermore, the identification accuracy of LYPJ was higher than those of the other four rice varieties, although the total recognition rate of 68.5% was not satisfactory. The results demonstrated that reflectance spectroscopy also showed a potential for monitoring the nitrogen status of hybrid rice.

## 3.3. Estimation of leaf chlorophyll content based on partial least square

The chlorophyll content of different varieties of hybrid rice was estimated by measuring the SPAD value using the hand-held SPAD instrument. The flag leaves of five plants from each zone were randomly

**Table 3.** Classification results of hybrid rice varieties for each nitrogen level based on the SVM model.

| | LYPJ | | YLY1 | | YLY2 | | YLY900 | | CY1000 | | total RCR |
|---|---|---|---|---|---|---|---|---|---|---|---|
| | RCR | RJR | RCR | RJR | RCR | RJR | RCR | RJR | RCR | RJR | |
| N1 | 100% | 100% | 71.4% | 100% | 87.5% | 100% | 100% | 90.3% | 100% | 100% | 91.8% |
| N2 | 100% | 100% | 71.4% | 80.7% | 50% | 93.3% | 28.6% | 93.6% | 100% | 100% | 70% |
| N3 | 87.5% | 100% | 100% | 93.6% | 50% | 100% | 85.7% | 100% | 100% | 86.7% | 84.6% |
| N4 | 100% | 100% | 100% | 100% | 100% | 90% | 85.7% | 93.6% | 37.5% | 96.7% | 84.6% |
| N5 | 100% | 100% | 100% | 100% | 75% | 96.7% | 85.7% | 93.6% | 100% | 100% | 92.1% |
| N6 | 50% | 96.7% | 85.7% | 90.3% | 87.5% | 86.7% | 42.9% | 96.8% | 62.5% | 96.7% | 65.7% |
| average | 89.6% | 99.5% | 88.1% | 94.1% | 75% | 94.5% | 71.4% | 94.6% | 83.3% | 96.7% | |

**Table 4.** Classification results of six nitrogen levels for each rice variety based on SVM model.

| | N1 | | N2 | | N3 | | N4 | | N5 | | N6 | | total RCR |
|---|---|---|---|---|---|---|---|---|---|---|---|---|---|
| | RCR | RJR | RCR | RJR | RCR | RJR | RCR | RJR | RCR | RJR | RCR | RJR | |
| LYPJ | 100% | 97.8% | 71.4% | 95.6% | 62.5% | 97.8% | 14.3% | 93.3% | 62.5% | 82.2% | 100% | 100% | 68.5% |
| YLY1 | 87.5% | 97.8% | 57.1% | 88.9% | 87.5% | 93.3% | 14.3% | 100% | 50% | 91.1% | 57.1% | 88.9% | 58.9% |
| YLY2 | 87.5% | 95.6% | 85.7% | 97.8% | 25% | 95.6% | 57.2% | 95.6% | 50% | 86.7% | 57.1% | 88.9% | 60.4% |
| YLY900 | 87.5% | 95.6% | 42.9% | 95.6% | 87.5% | 97.8% | 42.9% | 93.3% | 12.5% | 95.6% | 57.1% | 84.4% | 55.1% |
| CY1000 | 87.5% | 93.3% | 14.3% | 88.9% | 12.5% | 95.6% | 42.9% | 86.7% | 100% | 100% | 85.7% | 95.6% | 57.1% |
| average | 90% | 96% | 54.3% | 93.3% | 55% | 96% | 34.3% | 93.8% | 55% | 91.1% | 71.4% | 91.6% | |

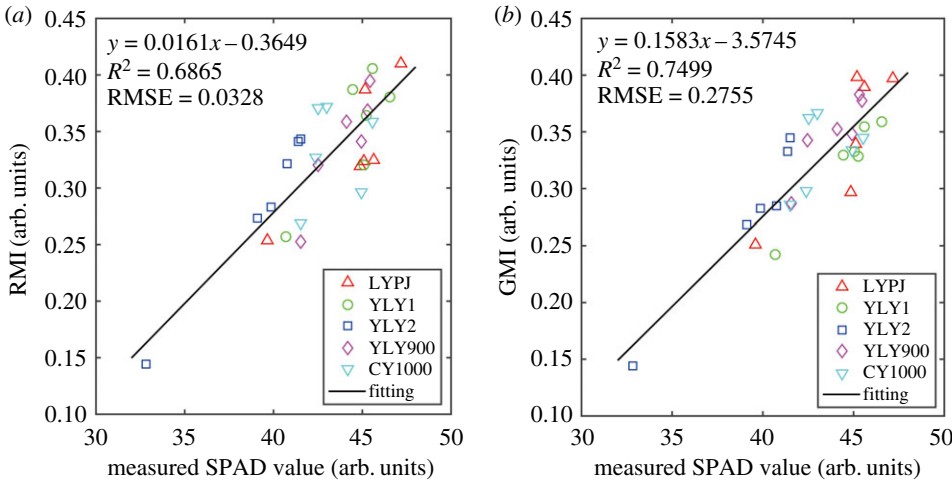

**Figure 2.** Relationship between spectral indices (RMI and GMI) and the SPAD value of different rice varieties.

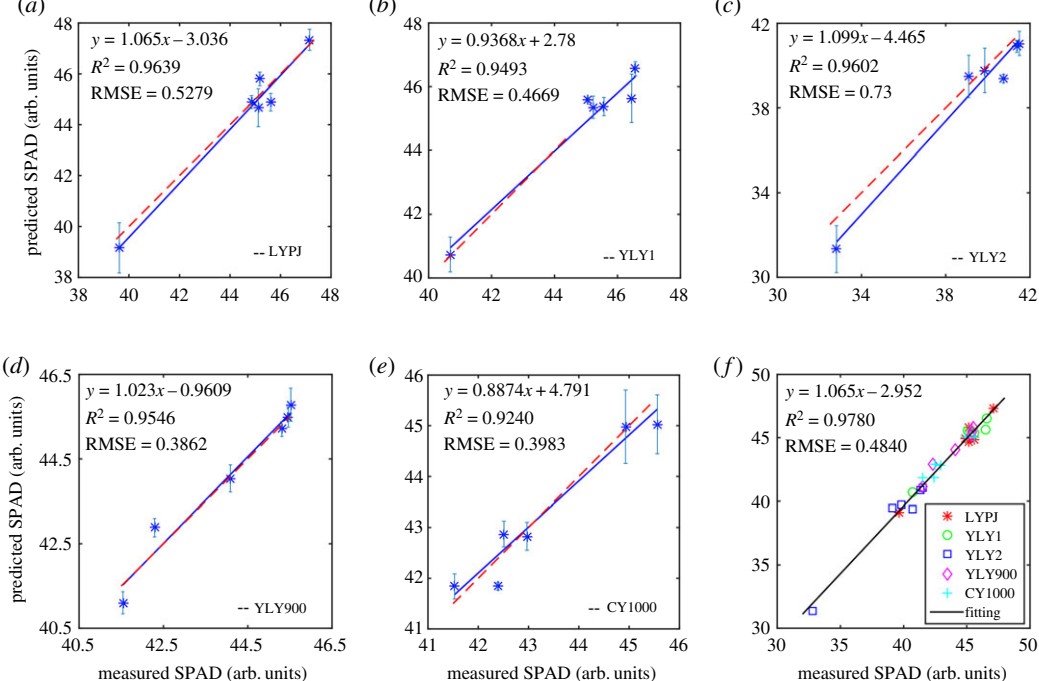

**Figure 3.** Relationship between measured and predicted SPAD based on 12 spectral indices. (*a*) LYP9, (*b*) LYL1, (*c*) LYL2, (*d*) YLY900, (*e*) CY1000 and (*f*) all rice varieties. The blue solid line represents the fitting curve, and the red dash line denotes the 1 : 1 line.

**Table 5.** Average and standard deviation of measured SPAD values.

|  | LYPJ | | YLY1 | | YLY2 | | YLY900 | | CY1000 | |
|---|---|---|---|---|---|---|---|---|---|---|
|  | average | s.d. | average | s.d. | average | s.d. | average | s.d. | average | s.d. |
| N1 | 39.62 | 1.49 | 40.69 | 0.84 | 32.81 | 4.16 | 41.54 | 2.90 | 41.53 | 1.39 |
| N2 | 44.85 | 2.10 | 45.07 | 2.02 | 39.84 | 3.93 | 42.49 | 1.59 | 42.39 | 1.48 |
| N3 | 45.12 | 1.77 | 45.23 | 2.49 | 39.09 | 2.55 | 44.09 | 2.89 | 42.51 | 2.24 |
| N4 | 45.61 | 2.13 | 45.57 | 1.91 | 40.74 | 2.34 | 45.31 | 1.04 | 42.97 | 2.99 |
| N5 | 45.18 | 2.23 | 46.57 | 1.24 | 41.37 | 0.87 | 45.46 | 2.41 | 45.56 | 5.69 |
| N6 | 47.15 | 0.99 | 44.45 | 1.74 | 41.5 | 2.36 | 45.52 | 3.07 | 44.93 | 2.56 |

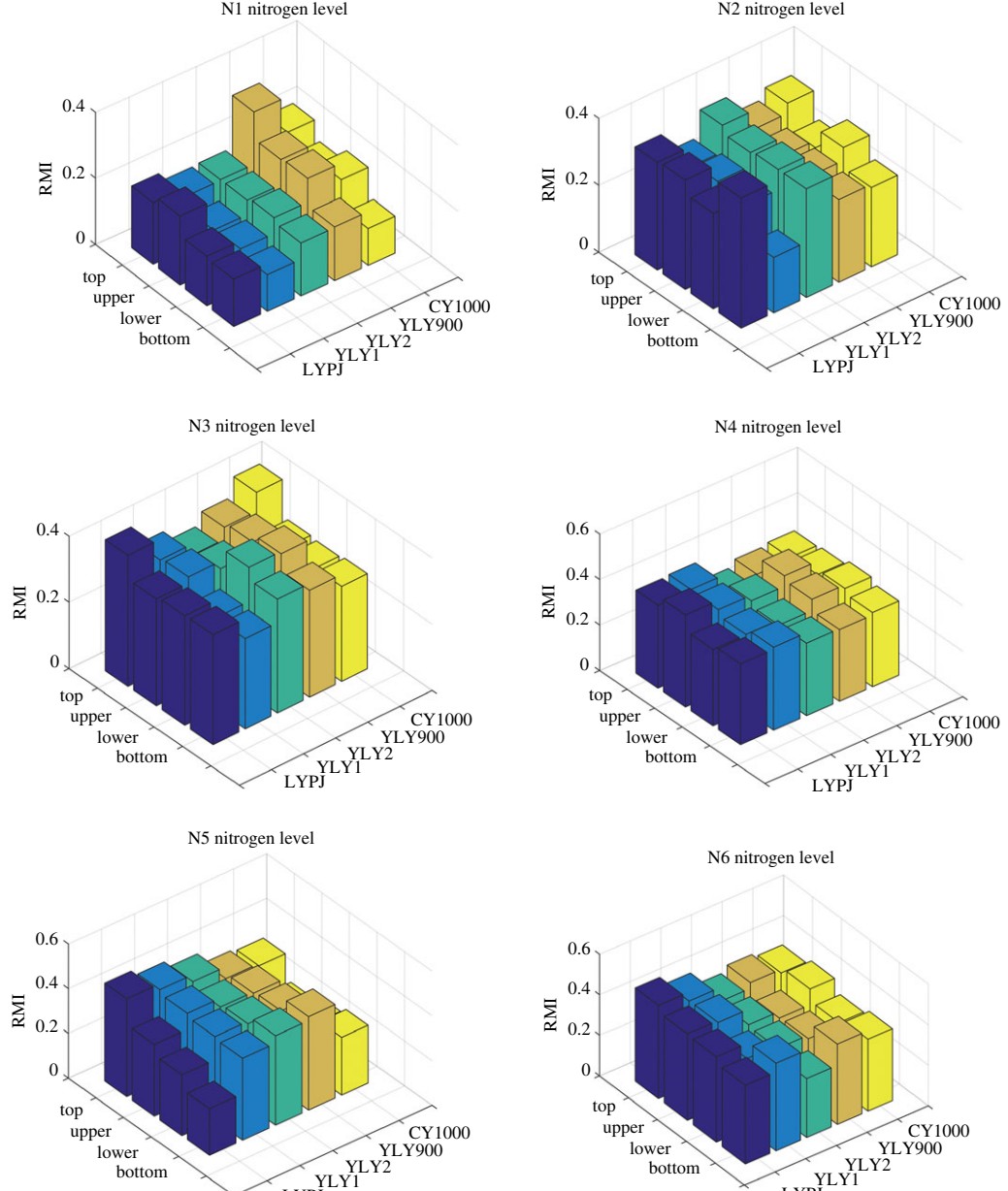

**Figure 4.** Vertical distribution of RMI for five rice varieties with six different nitrogen fertilizer levels.

selected and measured. The average and standard deviation of the measured SPAD values are shown in table 5. To test the feasibility of reflectance spectra for estimating chlorophyll content, two commonly used SIs, namely RMI and GMI, were first used to establish the correlation between the reflectance characteristics and the average SPAD value of different hybrid rice varieties, as shown in figure 2. Both RMI and GMI present a positive linearity with the SPAD values of all hybrid rice varieties, where the RMSE and $R^2$ were calculated as: $R^2 = 0.6865$, RMSE $= 0.033$ for RMI, and $R^2 = 0.7499$, RMSE $= 0.276$ for GMI. The results demonstrate that the relationship between the SIs and the average SPAD value was highly consistent for all the samples of different varieties of hybrid rice.

Furthermore, all 12 SIs described above were used to estimate SPAD values (or LCC) of hybrid rice. Figure 3a–e individually display the PLS regression results of the testing set for each rice variety, where mean values and error bars were used to characterize the distribution of the predicted SPAD values. The values of $R^2$ and RMSEP were calculated to evaluate the prediction performance of the PLS model. In figure 3, the blue solid lines represent the fitting curves and red dash lines represent the 1 : 1 lines. Obviously, the $R^2$ value of all rice varieties was more than 90% and the predicted SPAD values almost coincided with the 1 : 1 line. In contrast with YLY1 and CY1000, it seems that the predicted

SPAD values of YLY2 were far away from the 1 : 1 line, while the $R^2$ value was higher than that of these two varieties. This abnormal behaviour can be explained in terms of the fact that the RMSEP of YLY1 and CY1000 were smaller than that of YLY2. Figure 3$f$ shows the linear relationship between the measured and mean predicted SPAD of all hybrid rice varieties, with a preferable prediction accuracy of $R^2 =$ 0.978 and RMSE = 0.484. According to equations (2.2) and (2.5), for the PLS models from figure 3$a$–$f$, the RMSECV were calculated as 0.598, 0.550, 0.505, 0.266, 0.407 and 0.506, and the RPD values were calculated as 3.4, 3.7, 3.7, 6.0, 6.8 and 4.6, respectively. Clearly, the RMSECV values lower than 0.6, the $R^2$ values higher than 0.92 and the RPD values larger than 3.4 demonstrate that the PLS models show an excellent prediction ability for chlorophyll content. Besides, when compared to the results of RMI and GMI shown in figure 2, the combination of multiple spectral indices can significantly improve the prediction performance. Therefore, the SIs of vis/NIR reflectance spectra could be regarded as quite effective indicators for LCC estimations of hybrid rice.

## 3.4. Vertical distribution of chlorophyll content in a rice plant

To roughly characterize the vertical profile of LCC in an individual rice plant, the spectral measurements were also performed on four leaves taken from the position of the flag leaf to the bottom of the rice plant. The red-edge chlorophyll index RMI was calculated to quantitatively estimate the chlorophyll content. The RMI distribution results are shown in figure 4. On the whole, RMI presented a decreasing tendency from top to bottom in a rice plant, although abnormal behaviour occurred for some rice varieties, which was found to be consistent with the known vertical distribution of leaf nitrogen content [44,45]. Besides, with the increase of applied nitrogen levels, the chlorophyll content gradient was somewhat decreasing. This phenomenon may be caused by the ease by which nitrogen can redistribute. When nitrogen deficiency pertains, the nitrogen of old leaves transfers to new leaves, while sufficient nitrogen leads to a reduction of this effect. The results demonstrate that the vertical distribution of chlorophyll content can be used to reflect the nutrient status and nitrogen levels of rice plants, which can provide a great potential for improved diagnosis of the rice growing status and precision fertilization. Moreover, it can also be concluded that the highest chlorophyll content is found in the flag leaves during rice maturation, due to the fact that flag leaves are the last growing leaves. This is also the reason why SPAD measurements have often been performed on flag leaves to obtain the chlorophyll content of paddy rice in real applications.

## 4. Conclusion

In this study, reflectance spectral measurements coupled with multivariate analysis were shown to be quite powerful in identifying the rice varieties and evaluating chlorophyll content in hybrid rice under different nitrogen fertilizer levels. The SVM algorithm based on the reflectance spectrum ranging from 450–850 nm showed a better discrimination performance for identifying different rice varieties; however, it was found to be inadequate for different nitrogen fertilizer levels. Further, by selecting 12 appropriate SIs based on previous related studies, the PLS regression model was successfully designed and used to estimate the chlorophyll content or SPAD value of each rice variety, with an RMSECV of less than 0.6, an $R^2$ of more than 92% and an RPD value of more than 3.4 for each variety of hybrid rice. Then the vertical distribution in an individual rice crop was investigated, and the results indicated that the chlorophyll content in flag leaves is the highest. These findings further establish that it is possible to accurately evaluate the leaf chlorophyll content and nitrogen status, and further to manage applied nitrogen fertilizer of hybrid rice by using spectral reflectance. However, in this study, the hybrid rice samples were collected from only one location during one season, thus the established PLS model may impose some limitations for practical applications. In future, we are planning to select rice samples from different locations during different seasons to quantitatively estimate the influence of nitrogen application level on rice yield, rice quality and nitrogen utilization efficiency by combining reflectance and fluorescence spectroscopy. Moreover, more spectral indices and fluorescence parameters, as well as pattern recognition methods will be adopted to improve the prediction accuracy.

Data accessibility. Our data are available within the Dryad Digital Repository: http://dx.doi.org/10.5061/dryad.p8pq7fq [46].
Authors' contributions. H.Z. participated in the design of the study and drafted the manuscript; J.H., Q.Z. and S.S. contributed to conception and design, and helped draft the manuscript; Z.D., Y.L., G.Z., W.F., S.Z., T.P. and H.Z.

carried out the rice spectral measurement work; Z.D. and H.Z. carried out the statistical data and interpreted the data. All authors gave final approval for publication.

Competing interests. The authors declare there are no conflicts of interest.

Funding. This work was financially supported by the 948 Project of Ministry of Agriculture of China (grant no. 2015-Z45), the Guangdong Province Innovation Research Team Program (grant no. 201001D0104799318), the China Postdoctoral Science Foundation (grant no. 2017M612399), the Science and Technology Project of Henan Province (grant no. 182102110427) and the Science and Technology Innovation Project of Henan Agricultural University (grant no. KJCX2018A09).

Acknowledgements. The authors gratefully acknowledge the strong assistance by Zhao Chunli and the support of Prof. Sailing He.

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

**13**