## [Reviewer comments · Royal Society Open Science]

Review History

RSOS-190542.R0 (Original submission)

Review form: Reviewer 1

Is the manuscript scientifically sound in its present form?

No

Are the interpretations and conclusions justified by the results?

No

Is the language acceptable?

Yes

Is it clear how to access all supporting data?

No

Do you have any ethical concerns with this paper?

No

Have you any concerns about statistical analyses in this paper?

Yes

Recommendation?

Reject

Comments to the Author(s)

This manuscript describes a very simple experiment assessing rice leaf material based on nitrogen fertilizer treatments. The leaf material was then scanned using a Vis/NIR instrument. The spectral measurements were used to build calibrations for chlorophyll, and identify the rice varieties. I have several concerns about the current manuscript which make it difficult for me to support it moving forward in the review process.

Firstly the concept of using NIR to measure these traits is not new so I'm not sure where there is anything new to NIR science with this study.

The title suggests leaf nitrogen will be measured but I don't see anywhere its shown leaf nitrogen is measured, only that N fertilizer is used to give a range in chlorophyll. Thus the title needs to be changed to reflect the actual experiment. Also from I title it possible to understand the scientific question being asked. I wasn't sure what the scientific question was.

The samples were generated at one location in one year. It is known that the environment has an effect of spectral data and as such, the calibrations developed here, would probably only work on the same samples, grown at the same location. For any NIR calibrations to be considered practical, multiple growing environments and usually over different years provides the most robust models. At the least, a second experiment grown either at a different location or at the same location in 2017 would have provided a validation set. Currently, the experiment is using an internal validation. The robustness of a calibration is usually measuring by calculating the RPD (find Phil Williams references) . I couldn't calculate an RPD as there was no data presented for chlorophyll measurements other than the Figure. The actual data used to build a calibration should be reported with SD or SE for each measurement so the RMSE can be compared.

For Fig 4A, 4b and 4C, I think the low value is having a major influence on the R2 value.

Overall, I think this is a Proof of Concept experiment and could be published after corrections as a short communication.

Review form: Reviewer 2

Is the manuscript scientifically sound in its present form?

Yes

Are the interpretations and conclusions justified by the results?

Yes

Is the language acceptable?

No

Is it clear how to access all supporting data?

Yes

Do you have any ethical concerns with this paper?

No

Have you any concerns about statistical analyses in this paper?

No

Recommendation?

Major revision is needed (please make suggestions in comments)

Comments to the Author(s)

The manuscript of "Variety identification and nitrogen fertilizer detection of hybrid rice based on vis/NIR reflectance spectroscopy" presents an interesting finding that vis/NIR spectroscopy can have a great potential for identification of rice varieties and evaluation of nitrogen fertilizer levels. But I have some comments that might be worth considering.

- 1) Please provide the climatic and soil conditions for field planting.
- 2) In this article, authors use the reflectance spectroscopy to analyze the photosynthetic activity of paddy rice. Could you show the advantage of reflectance spectroscopy relative to other methods?
- 3) What are the figures used for from page 12 to page 16? And the figure number is missed.
- 4) The language should be carefully modified by a native speaker.
- 5) The format of references should be unified according to the instruments.

Review form: Reviewer 3

Is the manuscript scientifically sound in its present form?

Yes

Are the interpretations and conclusions justified by the results?

Yes

Is the language acceptable?

Yes

Is it clear how to access all supporting data?

Not Applicable

Do you have any ethical concerns with this paper?

No

Have you any concerns about statistical analyses in this paper?

No

Recommendation?

Major revision is needed (please make suggestions in comments)

Comments to the Author(s)

1. English of manuscript needs revision, there are few typos in the manuscript
2. I don't find the second objective of study clear, and I recommend to authors to revise it

3. Material and methods are written poorly and need extensive revision
 - 3.1. Authors should write the date after transplant for all stages, to make it easier to track
 - 3.2. Section 2.1: What was the total number of samples?
 - 3.3. Based on section 2.1 It is not clear spectral measurement been done in the field or lab, revise it to make it more clear
 - 3.4. How many reading for a sample performed? If several, Have you done the averaging?
 - 3.5. How authors, modified hyperparameters in SVM (i.e. Cost and Sigma)? It needs more elaboration!
 - 3.6. What is the ratio between training and testing of SVM?
4. Results and Discussion is mostly methodology which needs to be revised.
 - 4.1. Authors used only RM and GM for correlating with SPAD reading, but they included normalised difference indices and modified chlorophyll ratio index in section 3.3.1! Why haven't they been used?
 - 4.2. Authors missed any anticipation on results about vertical distribution of Chlorophyll content.
5. It would be good if authors provide a suggestion for future work

Decision letter (RSOS-190542.R0)

18-Jun-2019

Dear Dr Zhang:

Manuscript ID RSOS-190542 entitled "Variety identification and nitrogen fertilizer detection of hybrid rice based on vis/NIR reflectance spectroscopy" which you submitted to Royal Society Open Science, has been reviewed. The comments from reviewers are included at the bottom of this letter.

In view of the criticisms of the reviewers, the manuscript has been rejected in its current form. However, a new manuscript may be submitted which takes into consideration these comments.

Please note that resubmitting your manuscript does not guarantee eventual acceptance, and that your resubmission will be subject to peer review before a decision is made.

Your resubmitted manuscript should be submitted by 16-Dec-2019. If you are unable to submit by this date please contact the Editorial Office.

on behalf of Professor Luning Liu (Associate Editor) and Pietro Cicuta (Subject Editor)
openscience@royalsociety.org

Associate Editor Comments to Author (Professor Luning Liu):

Thank you for giving us the opportunity to review your manuscript for publication in Royal Society Open Science. Please find three reviewers' comments below. The reviewers have raised a number of concerns that must be addressed. As these concerns are of some gravity and will entail substantial changes to the manuscript, I must reject the manuscript in its present form. Nonetheless, if you are able to address the points raised by reviewers, I would welcome your resubmission of a manuscript that incorporates your responses.

Reviewers' Comments to Author:

Reviewer: 1

Comments to the Author(s)

This manuscript describes a very simple experiment assessing rice leaf material based on nitrogen fertilizer treatments. The leaf material was then scanned using a Vis/NIR instrument. The spectral measurements were used to build calibrations for chlorophyll, and identify the rice varieties. I have several concerns about the current manuscript which make it difficult for me to support it moving forward in the review process.

Firstly the concept of using NIR to measure these traits is not new so I'm not sure where there is anything new to NIR science with this study.

The title suggests leaf nitrogen will be measured but I don't see anywhere its shown leaf nitrogen is measured, only that N fertilizer is used to give a range in chlorophyll. Thus the title needs to be changed to reflect the actual experiment. Also from I title it possible to understand the scientific question being asked. I wasn't sure what the scientific question was.

The samples were generated at one location in one year. It is known that the environment has an effect of spectral data and as such, the calibrations developed here, would probably only work on the same samples, grown at the same location. For any NIR calibrations to be considered practical, multiple growing environments and usually over different years provides the most robust models. At the least, a second experiment grown either at a different location or at the same location in 2017 would have provided a validation set. Currently, the experiment is using an internal validation. The robustness of a calibration is usually measuring by calculating the RPD (find Phil Williams references) . I couldn't calculate an RPD as there was no data presented for chlorophyll measurements other than the Figure. The actual data used to build a calibration should be reported with SD or SE for each measurement so the RMSE can be compared.

For Fig 4A, 4b and 4C, I think the low value is having a major influence on the R2 value.

Overall, I think this is a Proof of Concept experiment and could be published after corrections as a short communication.

Reviewer: 2

Comments to the Author(s)

The manuscript of "Variety identification and nitrogen fertilizer detection of hybrid rice based on vis/NIR reflectance spectroscopy" presents an interesting finding that vis/NIR spectroscopy can

have a great potential for identification of rice varieties and evaluation of nitrogen fertilizer levels. But I have some comments that might be worth considering.

- 1) Please provide the climatic and soil conditions for field planting.
- 2) In this article, authors use the reflectance spectroscopy to analyze the photosynthetic activity of paddy rice. Could you show the advantage of reflectance spectroscopy relative to other methods?
- 3) What are the figures used for from page 12 to page 16? And the figure number is missed.
- 4) The language should be carefully modified by a native speaker.
- 5) The format of references should be unified according to the instruments.

Reviewer: 3

Comments to the Author(s)

1. English of manuscript needs revision, there are few typos in the manuscript
2. I don't find the second objective of study clear, and I recommend to authors to revise it
3. Material and methods are written poorly and need extensive revision
 - 3.1. Authors should write the date after transplant for all stages, to make it easier to track
 - 3.2. Section 2.1: What was the total number of samples?
 - 3.3. Based on section 2.1 It is not clear spectral measurement been done in the field or lab, revise it to make it more clear
 - 3.4. How many reading for a sample performed? If several, Have you done the averaging?
 - 3.5. How authors, modified hyperparameters in SVM (i.e. Cost and Sigma)? It needs more elaboration!
 - 3.6. What is the ratio between training and testing of SVM?
4. Results and Discussion is mostly methodology which needs to be revised.
 - 4.1. Authors used only RM and GM for correlating with SPAD reading, but they included normalised difference indices and modified chlorophyll ratio index in section 3.3.1! Why haven't they been used?
 - 4.2. Authors missed any anticipation on results about vertical distribution of Chlorophyll content.
5. It would be good if authors provide a suggestion for future work

Author's Response to Decision Letter for (RSOS-190542.R0)

See Appendix A.

RSOS-191132.R0

Review form: Reviewer 1

Is the manuscript scientifically sound in its present form?

No

Are the interpretations and conclusions justified by the results?

No

Is the language acceptable?

Yes

Do you have any ethical concerns with this paper?

No

Have you any concerns about statistical analyses in this paper?

No

Recommendation?

Major revision is needed (please make suggestions in comments)

Comments to the Author(s)

I believe the revisions have addressed most of the suggested corrections. However, I'm concerned the authors are making claims on the application of the calibrations based on one set of samples from one location in one season. There is no mention of this limitation in the abstract and as such, the abstract is misleading. It needs to be clearly stated that the data is from one season in one location. Further, there was no true validation with an independent validation set. The authors must acknowledge these major limitations in this study with no independent validation set which is the acceptable process when publishing NIR calibrations, and the authors must acknowledge their data is limited based on one season in one year. Hence any calibrations are limited to that set of rice grown at that location, in hopefully similar seasons. The authors must report a RMSECV in the abstract. Also this is not a RMSEP which is the true measure of the predicted error of calibration. Further, the authors presented a RPD which uses a standard deviation, but what was the standard deviation? Where's the data showing the mean and standard deviation of the X variable (SPAD units)? How many factors were used? Without all of this data, this cannot be published.

Review form: Reviewer 2

Is the manuscript scientifically sound in its present form?

Yes

Are the interpretations and conclusions justified by the results?

Yes

Is the language acceptable?

Yes

Do you have any ethical concerns with this paper?

No

Have you any concerns about statistical analyses in this paper?

No

Recommendation?

Accept as is

Comments to the Author(s)

The manuscript should be accepted as the present status.

Review form: Reviewer 3

Is the manuscript scientifically sound in its present form?

Yes

Are the interpretations and conclusions justified by the results?

Yes

Is the language acceptable?

No

Do you have any ethical concerns with this paper?

No

Have you any concerns about statistical analyses in this paper?

No

Recommendation?

Major revision is needed (please make suggestions in comments)

Comments to the Author(s)

Authors improved their manuscript but still needs more elaboration before it can be considered for publication:

0. English of the manuscript still needs improvement

Material and methods still need more elaboration, for instance:

1. Provide the full name of spectral indices.
2. What parameters used for Genetic algorithm, simply saying GA used for parameter selection is not enough and authors should elaborate more.
3. What is the main reason authors used Savitzky-Golay smoothing as a preprocessing method, more elaboration required?
4. Why authors used both R-square and RPD for evaluating their PLS model, both provide the same results.
5. Authors should provide the specification of their spectroradiometer.
6. What is the material of their standard reference? more elaboration here required.
- 7 Authors should explain How they record the dark reflectances
8. The authors should also add the number of Days after Transplantation or Plantation as well
9. Why didn't the authors use all 600 samples in their data processing?
10. How authors divide their samples into training and testing? Clarify it
11. Spectral measurements still are not clear? More elaboration is needed. Authors should note that more details such as position, angle and distance of the sensor to sample should be provided in the manuscript.

Decision letter (RSOS-191132.R0)

29-Jul-2019

Dear Dr Zhang,

The Subject Editor assigned to your paper ("Vis/NIR reflectance spectroscopy for hybrid rice

variety identification and chlorophyll content evaluation for different nitrogen fertilizer levels") has now received comments from reviewers. We would like you to revise your paper in accordance with the referee and Associate Editor suggestions which can be found below (not including confidential reports to the Editor). Please note this decision does not guarantee eventual acceptance.

Please submit a copy of your revised paper before 21-Aug-2019. Please note that the revision deadline will expire at 00.00am on this date. If we do not hear from you within this time then it will be assumed that the paper has been withdrawn. In exceptional circumstances, extensions may be possible if agreed with the Editorial Office in advance. We do not allow multiple rounds of revision so we urge you to make every effort to fully address all of the comments at this stage. If deemed necessary by the Editors, your manuscript will be sent back to one or more of the original reviewers for assessment. If the original reviewers are not available we may invite new reviewers.

When submitting your revised manuscript, you must respond to the comments made by the referees and upload a file "Response to Referees" in "Section 6 - File Upload". Please use this to document how you have responded to each of the comments, and the adjustments you have made. In order to expedite the processing of the revised manuscript, please be as specific as possible in your response.

- Ethics statement

- Data accessibility

<http://datadryad.org/submit?journalID=RSOS&manu=RSOS-191132>

- Competing interests

- Authors' contributions

- Acknowledgements

- Funding statement

on behalf of Professor Luning Liu (Associate Editor) and Pietro Cicuta (Subject Editor)
openscience@royalsociety.org

Reviewer comments to Author:

Reviewer: 2

Comments to the Author(s)

The manuscript should be accepted as the present status.

Reviewer: 1

Comments to the Author(s)

I believe the revisions have addressed most of the suggested corrections. However, I'm concerned the authors are making claims on the application of the calibrations based on one set of samples

from one location in one season. There is no mention of this limitation in the abstract and as such, the abstract is misleading. It needs to be clearly stated that the data is from one season in one location. Further, there was no true validation with an independent validation set. The authors must acknowledge these major limitations in this study with no independent validation set which is the acceptable process when publishing NIR calibrations, and the authors must acknowledge their data is limited based on one season in one year. Hence any calibrations are limited to that set of rice grown at that location, in hopefully similar seasons. The authors must report a RMSECV in the abstract. Also this is not a RMSEP which is the true measure of the predicted error of calibration. Further, the authors presented a RPD which uses a standard deviation, but what was the standard deviation? Where's the data showing the mean and standard deviation of the X variable (SPAD units)? How many factors were used? Without all of this data, this cannot be published.

Reviewer: 3

Comments to the Author(s)

Authors improved their manuscript but still needs more elaboration before it can be considered for publication:

0. English of the manuscript still needs improvement

Material and methods still need more elaboration, for instance:

1. Provide the full name of spectral indices.
2. What parameters used for Genetic algorithm, simply saying GA used for parameter selection is not enough and authors should elaborate more.
3. What is the main reason authors used Savitzky-Golay smoothing as a preprocessing method, more elaboration required?
4. Why authors used both R-square and RPD for evaluating their PLS model, both provide the same results.
5. Authors should provide the specification of their spectroradiometer.
6. What is the material of their standard reference? more elaboration here required.
- 7 Authors should explain How they record the dark reflectances
8. The authors should also add the number of Days after Transplantation or Plantation as well
9. Why didn't the authors use all 600 samples in their data processing?
10. How authors divide their samples into training and testing? Clarify it
11. Spectral measurements still are not clear? More elaboration is needed. Authors should note that more details such as position, angle and distance of the sensor to sample should be provided in the manuscript.

Editorial Office comments for the Authors:

For information about language-editing services endorsed by the Royal Society, please follow the link below:

<https://royalsociety.org/journals/authors/language-polishing/>

Author's Response to Decision Letter for (RSOS-191132.R0)

See Appendix B.

RSOS-191132.R1 (Revision)

Review form: Reviewer 2

Is the manuscript scientifically sound in its present form?

Yes

Are the interpretations and conclusions justified by the results?

Yes

Is the language acceptable?

Yes

Do you have any ethical concerns with this paper?

No

Have you any concerns about statistical analyses in this paper?

No

Recommendation?

Accept as is

Comments to the Author(s)

The manuscript should be accepted as the present status.

Review form: Reviewer 3

Is the manuscript scientifically sound in its present form?

Yes

Are the interpretations and conclusions justified by the results?

Yes

Is the language acceptable?

Yes

Do you have any ethical concerns with this paper?

No

Have you any concerns about statistical analyses in this paper?

No

Recommendation?

Accept as is

Comments to the Author(s)

Authors addressed all raised issues; thus, I can recommend it to be considered for publication.

Decision letter (RSOS-191132.R1)

24-Sep-2019

Dear Dr Zhang,

I am pleased to inform you that your manuscript entitled "Vis/NIR reflectance spectroscopy for hybrid rice variety identification and chlorophyll content evaluation for different nitrogen fertilizer levels" is now accepted for publication in Royal Society Open Science.

on behalf of Professor Luning Liu (Associate Editor) and Pietro Cicuta (Subject Editor)
openscience@royalsociety.org

Reviewer comments to Author:

Reviewer: 2

Comments to the Author(s)

The manuscript should be accepted as the present status.

Reviewer: 3

Comments to the Author(s)

Authors addressed all raised issues; thus, I can recommend it to be considered for publication.

Appendix A

The Editors,
Royal Society Open Science

Rebuttal letter; Hao Zhang et al., "Vis/NIR reflectance spectroscopy for hybrid rice variety identification and chlorophyll content evaluation for different nitrogen fertilizer levels"

Dear Editors,

Thank you for giving us the opportunity to resubmit our manuscript to the journal "Royal Society Open Science". The comments are very helpful and valuable. We have revised the manuscript carefully according to the reviewers' comments. The answers to the comments are as follows:

Reviewer: 1

Point 1: "Firstly the concept of using NIR to measure these traits is not new so I'm not sure where there is anything new to NIR science with this study."

Response 1: We admit that many related studies have been performed, and we have referred to many such papers in our Introduction. However, in our work, we mainly focused on the measurement of chlorophyll content in hybrid rice under different fertilizer levels, an area where only little previous work has been performed.

Point 2: "The title suggests leaf nitrogen will be measured but I don't see anywhere its shown leaf nitrogen is measured, only that N fertilizer is used to give a range in chlorophyll. Thus the title needs to be changed to reflect the actual experiment. Also from I title it possible to understand the scientific question being asked. I wasn't sure what the scientific question was."

Response 2: We recognized that the title is not suitable to reflect the objective of the studies. In this work, we mainly focused on (1) identify the hybrid rice varieties and the nitrogen fertilization levels by using the SVM method; (2) analyze the performance of selected spectral indices (SIs) and the partial least square (PLS) method for the estimation of chlorophyll content (or SPAD value), so we have changed the title to the more appropriate "Vis/NIR reflectance spectroscopy for hybrid rice variety identification and chlorophyll content evaluation for different nitrogen fertilizer levels".

Point 3: "The samples were generated at one location in one year. It is known that the environment has an effect of spectral data and as such, the calibrations developed here, would probably only work on the same samples, grown at the same location. For any NIR calibrations to be considered practical, multiple growing environments and usually over different years provides the most robust models. At the least, a second experiment grown either at a different location or at the same location in 2017 would have provided a validation set. Currently, the experiment is using an internal validation. The robustness of a calibration is usually measuring by calculating the RPD (find Phil Williams references). I couldn't calculate an RPD as there was no data

presented for chlorophyll measurements other than the Figure. The actual data used to build a calibration should be reported with SD or SE for each measurement so the RMSE can be compared.”

Response 3: For NIR calibrations, we have only evaluated the PLS model with an internal validation, by using the root mean square error (RMSE) of the prediction and the coefficient of determination (R^2). According to the reviewer’s suggestion, we added the ratio of performance to deviation (RPD) to further validate the performance of a PLS model. The standard deviations (SD) of the measured SPAD values and the RMSE of PLS model were used to calculate the RPD values. The calculated RPD values found to be higher than 3.4 demonstrated that the PLS models show an excellent prediction ability for chlorophyll content. The added sentences can be found on page 4 and page 6 of the revised manuscript.

Point 4: “For Fig 4A, 4b and 4C, I think the low value is having a major influence on the R^2 value.”

Response 4: We agree, the observation is correct. The low SPAD value has a major influence on the prediction accuracy of the PLS model.

Reviewer: 2

Point 1: “Please provide the climatic and soil conditions for field planting.”

Response 1: We have provided the climatic and soil conditions on page 2 of the revised manuscript, “Xinyang is located in the transition zone between the northern subtropical to warm temperate zones, with the climatic conditions of enough summer sunlight, high temperature, and abundant rain, which constitute ideal conditions for rice planting. The soil is a typical paddy soil, which is one of the major farming types in China.”.

Point 2: “In this article, authors use the reflectance spectroscopy to analyze the photosynthetic activity of paddy rice. Could you show the advantage of reflectance spectroscopy relative to other methods?”

Response 2: We have provided the advantages of reflectance spectroscopy on page 2 of the manuscript, “Compared to the LIF technique, reflectance spectroscopy has the advantages of compact and cheap instrumentation. ”.

Point 3: “What are the figures used for from page 12 to page 16? And the figure number is missed.”

Response 3: The total pages of our manuscript are 12, so we cannot find this problem. Besides, we also checked the figure numbers - they are correct so there is some misunderstanding by the referee.

Point 4: “The language should be carefully modified by a native speaker.”

Response 4: The language has been modified by a native speaker.

Point 5: “The format of references should be unified according to the instruments.”

Response 5: The format of references has been revised according to the instructions.

Reviewer: 3

Point 1: “English of manuscript needs revision, there are few typos in the manuscript”

Response 1: The English has been modified carefully by a native speaker.

Point 2: “I don't find the second objective of study clear, and I recommend to authors to revise it”

Response 2: The title has been changed to “Vis/NIR reflectance spectroscopy for hybrid rice variety identification and chlorophyll content evaluation for different nitrogen fertilizer levels”. With the new title the two objectives of the study should now be clear.

Point 3: “Material and methods are written poorly and need extensive revision”

Response 3: We have re-worked the section of “Materials and Methods”.

Point 4: “Authors should write the date after transplant for all stages, to make it easier to track”

Response 4: We have added the date after transplant for all stages, please find it on page 3 of the revised manuscript.

Point 5: “Section 2.1: What was the total number of samples?”

Response 5: For each zone, five whole plants were picked from the field, and thus totally 150 whole plants were collected. For each plant, 4 leaf samples were selected by sampling from its top (flag leaf) to its bottom. For each leaf, 9 different positions were selected for spectral measurements. Thus, the total number of leaf samples was 600. We have added this information on page 3 of the revised manuscript, “Thus, the total number of leaf samples for spectral measurements was 600.”

Point 6: “Based on section 2.1 It is not clear spectral measurement been done in the field or lab, revise it to make it more clear”

Response 6: The spectral measurement was done in the mobile laboratory nearby the field, we have re-written the sentence to make it more clear; please find it in page 3 of the revised manuscript, “The spectral measurements were carried out in a newly constructed mobile laboratory from South China Normal University as described in [28]. The mobile laboratory was positioned nearby the rice planting fields to facilitate the reflectance measurements.”.

Point 7: “How many reading for a sample performed? If several, Have you done the averaging?”

Response 7: During spectral measurements, each sample was repeatedly measured three times, and the recorded data were averaged to a resulting spectrum. We have added the information on page 3 of the revised manuscript, “Each leaf sample was repeatedly measured three times and the recorded data were then averaged.”.

Point 8: “How authors, modified hyperparameters in SVM (i.e. Cost and Sigma)? It needs more elaboration!”

Response 8: We have added the detailed description for the optimization of the SVM model, including the selection of the parameters C and γ . On page 4 of the revised manuscript: “The penalty parameter C and kernel function parameter γ play a very important role in controlling the modelling complexity and classification accuracy of the SVM model. Therefore, in this work, a genetigorithm (GA) was utilized to select appropriate parametec alrs C and γ to obtain an optimized SVM model, where the best

C and γ were determined according to the best classification accuracy based on K-fold cross validation (K-CV).”.

Point 9: “What is the ratio between training and testing of SVM?”

Response 9: The ratio between training and testing of SVM was about 5:1. We have added it on page 5 of the revised manuscript, “With a number ratio of 5:1, the spectral data were divided into a training set and a testing set,”.

Point 10: “Results and Discussion is mostly methodology which needs to be revised.”

Response 10: We have re-arranged the section of “Results and Discussion”, the part of “Selection of spectral indices” was moved to the section of “Materials and Methods”, and we also added some discussions.

Point 11: “Authors used only RM and GM for correlating with SPAD reading, but they included normalised difference indices and modified chlorophyll ratio index in section 3.3.1! Why haven't they been used? ”

Response 11: In this work, we selected 12 kinds of spectral indices for analyzing the chlorophyll content. We first only used RM and GM to validate the correlation relationship between reflectance spectra and SPAD values. Subsequently, we used all 12 spectral indices to establish the PLS regression model to find the correlation relationship between reflectance spectra and SPAD values. By contrast, the combination of multiple spectral indices can greatly improve the prediction performance.

Point 12: “Authors missed any anticipation on results about vertical distribution of Chlorophyll content.”

Response 12: We have added some discussions about vertical distribution of chlorophyll content, please find it on page 6 of the revised manuscript: “.... which is consistent with the known vertical distribution of leaf nitrogen content [44, 45]. Besides, with the increase of nitrogen application levels, the chlorophyll content gradient is somewhat decreasing. This phenomenon may be caused by the ease by which nitrogen can redistribute. When nitrogen deficiency pertains, the nitrogen of old leaves transfer to new leaves, while sufficient nitrogen leads to a reduction of this effect. The results demonstrate that the vertical distribution of chlorophyll content can be used to reflect the nutrient status and nitrogen levels of rice plants, which could provide a great potential for improved diagnosis of the rice growing status and precision fertilization. ”.

Point 13: “It would be good if authors provide a suggestion for future work”

Response 13: As the reviewer suggested, we have added the future work in page 6-7 of the revised manuscript, “For the future, we are planning to quantitatively estimate the influence of nitrogen application level on rice yield, rice quality, and nitrogen utilization efficiency by combining reflectance and fluorescence spectroscopy. In addition, more spectral indices and fluorescence parameters, as well as pattern recognition methods could be adopted to improve the prediction accuracy.”

With our reworked manuscript, where the reviewers' points have been thoroughly considered, we hope that you will now find our paper suitable for publication in “Royal Society Open Science”.

For all the authors,

Yours Sincerely,

Hao Zhang

Appendix B

The Editors,
Royal Society Open Science

Rebuttal letter; Hao Zhang et al., "Vis/NIR reflectance spectroscopy for hybrid rice variety identification and chlorophyll content evaluation for different nitrogen fertilizer levels"

Dear Editors,

Thank you for your letter and the reviewers' comments concerning our manuscript. The comments are very helpful and valuable for improving our manuscript. We have read the comments carefully and made corrections. The revised portions are marked in red in the revised manuscript. The responds to the reviewer's comments are as follows, where our own comments are marked in blue:

Reviewer: 1

Point 1: "I'm concerned the authors are making claims on the application of the calibrations based on one set of samples from one location in one season. There is no mentioned of this limitation in the abstract and as such, the abstract is misleading. It needs to be clearly stated that the data is from one season in one location. Further, there was no true validation with an independent validation set. The authors must acknowledge these major limitations in this study with no independent validation set which is the acceptable process when publishing NIR calibrations, and the authors must acknowledge their data is limited based on one season in one year. Hence any calibrations are limited to that set of rice grown at that location, in hopefully similar seasons."

Response 1: We admitted that we have performed the spectral measurements on only one season and one location, indeed, the prediction performance of the established PLS model would be influenced by the climatic and geographical conditions. Therefore, as the Reviewer mentioned, the PLS calibration has limitations. According to the Reviewer's suggestion, we have now, as requested added "....., five hybrid rice varieties were cultivated during one whole growing period in one experimental field supplied with six nitrogen fertilizer levels." to the abstract. As well, we have described the limitations of this study and the further work to improve it in the conclusion part, "However, in this study, the hybrid rice samples were collected from only one location during one season, thus the established PLS model may impose some limitations for practical applications. In future, we are planning to select rice samples from different locations during different seasons to quantitatively estimate the influence of nitrogen application level on rice yield, rice quality, and nitrogen utilization efficiency by combining reflectance and fluorescence spectroscopy."

Point 2: "The authors must report a RMSECV in the abstract. Also this is not a RMSEP which is the true measure of the predicted error of calibration."

Response 2: We have added the calculation equation of RMSECV in section

“Materials and Methods”, and also calculated the RMSECV values in section “Results and Discussion”. In the abstract, we added the RMSECV value, “, and a root mean square error of cross-validation (RMSECV) of 0.506,”.

Point 3: “ Further, the authors presented a RPD which uses a standard deviation, but what was the standard deviation? Where's the data showing the mean and standard deviation of the X variable (SPAD units)? How many factors were used? Without all of this data, this cannot be published.”

Response 3: We have provided the data showing the mean and standard deviation of measured SPAD values, shown in Table 5.

Reviewer: 2

The manuscript should be accepted as the present status.

Response: We appreciate this positive comment.

Reviewer: 3

Point 1: “English of the manuscript still needs improvement”

Response 1: We have now further improved the linguistic aspects with revisions marked in red in the revised manuscript.

Point 2: “Provide the full name of spectral indices.”

Response 2: We have provided the full name of all spectral indices in the section “ 2.3 Selection of spectral indices”, including simple ratio (SR), red-edge model index (RM), green-edge model index (GM), normalized difference vegetation index (NDVI), modified chlorophyll absorption ratio index (MCARI), triangular vegetation index (TVI), and modified triangular vegetation index (MTVI).

Point 3: “What parameters used for Genetic algorithm, simply saying GA used for parameter selection is not enough and authors should elaborate more.”

Response 3: We have added a sentence to describe the parameters used for the genetic algorithm, “..... where the parameters used for GA were set as: maximum generation of 100, population size of 20, and crossover probability of 0.9. The”.

Point 4: “What is the main reason authors used Savitzky-Golay smoothing as a preprocessing method, more elaboration required?”

Response 4: We have now described the reason in detail, “During spectral measurements, owing to the influence of non-ideal instruments and sample properties, the obtained reflectance spectra were much noisy, which introduced some large errors to the estimation of chlorophyll content when using the SIs. Therefore, to reduce the noise and increase the signal to noise ratio, the reflectance spectra were processed by Savitzky–Golay (SG) smoothing method, with a third order polynomial approximation and a window size of 10 points.”.

Point 5: “Why authors used both R-square and RPD for evaluating their PLS model, both provide the same results.”

Response 5: Generally, the commonly used indicators for evaluating the performance of regression model consist of the root mean square error of cross-validation (RMSECV), the root mean square error of prediction (RMSEP), the coefficient of

determination (R^2), and the ratio of performance to deviation (RPD), where RMSECV, RMSEP, and R^2 belongs to internal cross validation. Thus, we selected both R^2 and RPD values to evaluate the performance of the PLS model, making the results more convincing. In addition, another reviewer also suggested us to calculate the RPD values.

Point 6: “Authors should provide the specification of their spectroradiometer.”

Response 6: We have now provided the specification of the SPAD instrument, “.....using the hand-held SPAD meter (SPAD-502Plus).”.

Point 7: “What is the material of their standard reference? more elaboration here required.”

Response 7: We have added further description of the standard reference, “.....a standard reference made with polytetrafluoroethylene material (the reflectivity is more than 98% in the wavelength range of 250–2200 nm).”.

Point 8: “Authors should explain How they record the dark reflectances”

Response 8: We have now provided the recording procedure of dark reflectances, “The dark reflectance spectra were recorded when all lamps in the laboratory were turned off and the fibers were obscured with dark papers.”.

Point 9: “The authors should also add the number of Days after Transplantation or Plantation as well”

Response 9: We have added all the date after transplantation of rice, “....., and 40% as panicle fertilizer at the 4th and 2nd leaf-age (June 25–July 4, 2016).”, “.....and 50% as spikelet-promoting fertilizer at 2nd leaf-age (July 2–4, 2016).”.

Point 10: “Why didn't the authors use all 600 samples in their data processing?”

Response 10: During the measurements, we totally used 600 samples, including different varieties and nitrogen levels, as well as different positions. In fact, for different research purposes, the used number of samples was different. When we used SVM model for identification of rice varieties, we only selected the samples from the same nitrogen level and the same position (i.e. the flag leaf); the number of total samples is 225. When we used the SVM model for identification of nitrogen levels, we only selected the samples from the same variety and the same position (i.e. the flag leaf); the number of total samples is 270. When we used PLS for estimation of leaf chlorophyll content, for each variety, the number of total samples was also 270. At the end, when we investigated the vertical distribution of chlorophyll content, we also selected 270 samples from different varieties and different nitrogen levels.

Point 11: “How authors divide their samples into training and testing? Clarify it ”

Response 11: We have described the procedure how to divide the samples into training and testing set in the section “3.2 Identification of rice varieties and nitrogen levels based on SVM”, “A total of 225 reflectance spectra from five varieties of hybrid rice were used for classification. With a number ratio of 5:1, the spectral data were divided into a training set and a testing set, where 187 spectra among them were used as training set and the remaining 38 spectra were used as testing set.”, “Then, the SVM method was also applied to identify different levels of nitrogen fertilizer. A total of 270 reflectance spectra from each variety of hybrid rice were used for classification. Similarly, with a number ratio of 5:1, the spectral data were divided into a training set

and a testing set, where 225 spectra among them were used as training set and the remaining 45 spectra were used as testing set.”.

Point 12: “ Spectral measurements still are not clear? More elaboration is needed. Authors should note that more details such as position, angle and distance of the sensor to sample should be provided in the manuscript.”

Response 12: We have provided more description about spectral measurements, “, the broadband light was guided with a 600 μm core-diameter fiber (transmitting fiber) to vertically irradiate the sample, where the distance between the fiber end facet and the sample was about 5 mm. The diffuse reflection light was captured using another fiber (collecting fiber) with core diameter of 600 μm and fixed at an angle of 45° , where the fiber end was placed at the same height as the transmitting fiber,”.

We have now further improved our manuscript as requested. The changes are marked in red in the revised manuscript. We much appreciate for Editors/Reviewers' kind work earnestly, and hope that our actions will meet with approval. Once again, thank you very much for your comments and suggestions.

For all the authors,

Yours Sincerely,

Hao Zhang